# ADVERSARIAL TRAINING USING CONTRASTIVE DIVERGENCE

## ABSTRACT

To protect the security of machine learning models against adversarial examples, adversarial training becomes the most popular and powerful strategy against various adversarial attacks by injecting adversarial examples into training data. However, it is time-consuming and requires high computation complexity to generate suitable adversarial examples for ensuring the robustness of models, which impedes the spread and application of adversarial training. In this work, we reformulate adversarial training as a combination of stationary distribution exploring, sampling, and training. Each updating of parameters of DNN is based on several transitions from the data samples as the initial states in a Hamiltonian system. Inspired by our new paradigm, we design a new generative method for adversarial training by using Contrastive Divergence (ATCD), which approaches the equilibrium distribution of adversarial examples with only few iterations by building from small modifications of the standard Contrastive Divergence (CD). Our adversarial training algorithm achieves much higher robustness than any other state-of-the-art adversarial training acceleration method on the ImageNet, CIFAR-10, and MNIST datasets and reaches a balance between performance and efficiency.

## 1 INTRODUCTION

Although deep neural networks have become increasingly popular and successful in many machine learning tasks (e.g., image recognition He et al. (2016b), speech recognition Hinton et al. (2012); van den Oord et al. (2016) and natural language processing Hochreiter & Schmidhuber (1997); Vaswani et al. (2017)), the discovery of adversarial examples Szegedy et al. (2014); Goodfellow et al. (2015) has attracted great attention to strengthening the robustness of deep neural network (DNN) under such subtle but malicious perturbations. These crafted samples pose potential security threats in various safety-critical tasks such as autonomous vehicles Evtimov et al. (2017) or face recognition Sharif et al. (2016); Dong et al. (2019), which are required to be highly stable and reliable.

Unfortunately, it is considered to be unresolved since no final conclusion has yet been reached on the root of the adversarial examples. Many defense methods Papernot et al. (2016); Na et al. (2018); Buckman et al. (2018) motivated by different interpretability of adversarial examples Goodfellow et al. (2015); Fawzi et al. (2018); Ma et al. (2018) were broken within a short time, indicating that there is still no thorough solution to settle this matter once and away. Nonetheless, adversarial training Szegedy et al. (2014); Goodfellow et al. (2015) has shown its ability to make classifiers more robust against sorts of attacks than any other defenses in Madry et al. (2018); Athalye et al. (2018). It offers an intuitive approach to handle the problem, which first obtains suitable adversarial examples by solving the inner maximization problem and then update the parameters of ML model from these examples by outer minimization. More and more advanced defenses Kannan et al. (2018); Lin et al. (2019); Xie et al. (2019); Zhang et al. (2019c) are developed based on adversarial training.

However, a major issue of the current adversarial training methods is their significantly higher computational cost than regular training. It often needs multiple days and hundreds of GPUs for ImageNet-like datasets to achieve better convergence Xie et al. (2019), which makes it nearly intractable and impractical for large models on tons of data. Even for small-sized datasets like CIFAR10, adversarial training takes much longer time than regular training.

To address this issue, we formulate the problem of generating adversarial examples in a Hamiltonian Monte Carlo framework (HMC) Neal et al. (2011), which can be considered as exploring the

stationary distribution of adversarial examples for current parameters. The high computational cost of adversarial training can be easily attributed to the long trajectory of HMC producing. Therefore, we propose a new adversarial training algorithm called ATCD for strengthening the robustness of target models, enlightened by the Contrastive Divergence (CD) Hinton (2002). We minimize the difference of Kullback-Leibler divergence between two adjacent sampling steps to avoid running long Monte-Carlo Markov Chains (MCMC). Instead of running the chain to achieve equilibrium, we can simply run the chain for fewer or even only one full step and then update the parameters to reduce the tendency of the chain to wander away from the initial distribution on the first step. Our approach is advantageous over existing ones in three folds:

- We offer a new perspective on adversarial examples generation in a HMC framework. From the view of HMC, we bridge the relationship between several adversarial examples generating methods and MCMC sampling, which effectively draw multiple fair samples from the underlying distribution of adversarial examples.

- By analyzing the trajectory shift of different lengths of MCMC simulating, we speed up the adversarial training by proposing a contrastive adversarial training (ATCD) method, which accelerates the process of achieving distribution equilibrium.

- We thoroughly compare the effectiveness of our algorithm in various settings and different architectures on ImageNet, CIFAR10 and MNIST. Models trained by our proposed algorithm achieve robust accuracies markedly exceeding the ones trained by regular adversarial training and the state-of-the-art speedup methods when defending against several attacks.

## 2    BACKGROUND AND RELATED WORK

**Adversarial Defense.** To deal with the threat of adversarial examples, different strategies have been studied to find countermeasures to protect ML models. These approaches can be roughly categorized into two main types: (a) detection only and (b) complete defense. The former approaches Bhagoji et al. (2018); Ma et al. (2018); Lee et al. (2018); Tao et al. (2018); Zhang et al. (2018) is to reject the potential malignant samples before feeding them to the ML models. The latter defenses obfuscate the gradient information of the classifiers to confuse the attack mechanisms including gradient masking Papernot & McDaniel (2017); Athalye et al. (2018) or randomized models Liu et al. (2018); Xie et al. (2018a); Lecuyer et al. (2019); Liu et al. (2019). There are also some add-ons modules Xie et al. (2019); Svoboda et al. (2019); Akhtar et al. (2018); Liao et al. (2018) being appended to the targeted network or adversarial interpolation schemes Zhang & Xu (2020); Lee et al. (2020) to protect deep networks against the adversarial attacks.

**Fast Adversarial Training.** Besides all the above methods, adversarial training Goodfellow et al. (2015); Kurakin et al. (2017); Kannan et al. (2018); Madry et al. (2018); Tramèr et al. (2018); Liu & Hsieh (2019); Wang et al. (2020; 2019) is the most effective way to ensure better robustness, which has been widely verified in many works and competitions. However, limited works focus on boosting robust accuracy with reasonable training speed. Free Shafahi et al. (2019) recycle the gradient information computed to reduce the overhead cost of adversarial training. YOPO Zhang et al. (2019b) recast the adversarial training as a discrete time differential game and derive a Pontryagin's Maximum Principle (PMP) for it. Fast-FGSM Wong et al. (2020) combines FGSM with random initialization to accelerate the whole process.

**Markov Chain Monte Carlo Methods.** Markov chain Monte Carlo (MCMC) Neal (1993) provides a powerful framework for exploring the complex solution space and achieves a nearly global optimal solution independent of the initial state. But the slow convergence rate of MCMC hinders its wide use in time critical fields. By utilizing the gradient information in the target solution space, Hamiltonian (or Hybrid) Monte Carlo method (HMC) Duane et al. (1987); Neal et al. (2011) achieves tremendous speed-up in comparison to previous MCMC algorithms. Multiple variants of HMC Pasarica & Gelman (2010); Salimans et al. (2015); Hoffman & Gelman (2014) were yet to be developed for adaptively tuning step size or iterations of leapfrog integrator. The fusion of MCMC and machine learning Tu & Zhu (2002); Chen et al. (2014); Song et al. (2017); Xie et al. (2018b) also shows great potential of MCMC.

**Contrastive Divergence.** Contrastive Divergence (CD) has achieved notable success in training energy-based models including Restricted Boltzmann Machines (RBMs) as an efficient training

method. The standard approach to estimating the derivative of the log-likelihood function is using the Markov chain Monte Carlo Gilks et al. (1995), which can be expressed as the difference of two expectations. It runs $k$ MCMC transition steps at each iteration $T$ and iteratively generates a sequence of parameter estimates $\{\theta_T\}_{T \geq 0}$ given an i.i.d. data sample $\{X_i\}_{i=1}^{N} \sim p_{\bar{\theta}}$, where $p_{\bar{\theta}}$ is the distribution of target samples for the true parameter $\bar{\theta}$. To reduce the computational complexity, the traditional Contrastive Divergence algorithm computes approximate RBM log-likelihood gradient setting $k = 1$. Various works are devoted to addressing the problem of the vanilla CD afterwards, such as uncontrolled biases and divergence Carreira-Perpinan & Hinton (2005); Yuille (2005); MacKay (2001); Fischer & Igel (2011; 2014). Persistent CD (PCD) and its relevant works Tieleman (2008); Tieleman & Hinton (2009); Desjardins et al. (2010) show a steady decrease of the log-likelihood in many numerical analysis while some works Schulz et al. (2010); Fischer & Igel (2010) also give examples in which PCD failed to converge. Although none of these works provide a solid convergence guarantee since the major problems of CD family stem from the fact that the stochastic approximation to the true gradient is a biased estimator, our work do not need the exact values of the derivatives. Actually, we just borrow the idea from the vanilla CD to accelerate the process of distribution equilibrium over the visible variables instead of discovering the unknown distribution Pang et al. (2018); Alayrac et al. (2019).

## 3 PRELIMINARIES

Considering a target DNN model $\hat{f} \in \mathcal{F}$, where $\mathcal{F}$ is the solution function space for classification task. We assume softmax is employed for the output layer of the model $f(\cdot)$ and let $f(x)$ denote the softmax output of a given input $x \in \mathbb{R}^d$, i.e., $f(x) : \mathbb{R}^d \to \mathbb{R}^C$, where $C$ is the number of categories. We also assume that there exists an oracle mapping function $f^* \in \mathcal{F} : x \mapsto y^*$, which pinpoints the belonging of the input $x$ to all the categories by accurate confidence scores $y^* \in \mathbb{R}^C$. The common training is to minimize the cross-entropy (CE) loss $J_{ce}$, which is defined as:

$$f = \arg\min_{f \in \mathcal{F}} \quad \mathbb{E}_{(x,y) \sim \mathcal{D}} \left[ J_{ce} \left( f(x), y \right) \right],$$  (1)

where $y$ is the manual one-hot annotation of the input $x$ since $y^*$ is invisible. The goal of Eq. (1) is to update the parameters of $f$ for better approaching $f^*$, which leads to $f(x) \approx y \approx y^* = f^*(x)$. Suppose the target DNN model correctly classifies most of the input after hundreds of iterations, it will still be badly misclassified by adversarial examples (i.e., $\arg\max_{c \in \{1, \cdots, C\}} f(\tilde{x})_c \neq y[c]$). In adversarial training, these constructed adversarial examples are used to updates the model using minibatch SGD. The objective of this minmax game can be formulated as a robust optimization following Madry et al. (2018):

$$f' = \arg\min_{f \in \mathcal{F}} \mathbb{E}_{(x,y) \sim \mathcal{D}} \left[ \max_{\tilde{x} \in \mathcal{N}(x)} J_{ce} \left( f(\tilde{x}), y \right) \right],$$  (2)

where the inner maximization problem attempts to generate the most easily misclassified samples while the outer minimization problem is to search a mapping function $f'$ which is the closest one to the oracle $f^*$.

## 4 HAMILTONIAN MONTE CARLO FOR ADVERSARIAL LEARNING

### 4.1 AN OVERVIEW OF MCMC AND HAMILTONIAN MONTE CARLO

The crux of this work relies on offering a fundamentally different view of adversarial example generation, which simulates the inner maximization in Eq. (2) as proposing dynamics by HMC. We now give the overall description of Metropolis-Hasting based MCMC algorithm. Suppose $p$ is our target distribution over a space $\mathcal{D}$, MCMC methods construct a Markov Chain that has the desired distribution $p$ as its stationary distribution. At the first step, MCMC chooses an arbitrary point $x_0$ as the initial state. Then it repeatedly performs the dynamic process consisting of the following steps: (1) Generate a candidate sample $\tilde{x}$ as a "proposed" value for state $x_{t+1}$ from the candidate-generating density $Q(x_t|\tilde{x})$. (2) Compute the acceptance probability $\xi = \min(1, \frac{p(\tilde{x})Q(x_t|\tilde{x})}{p(x_t)Q(\tilde{x}|x_t)})$, which is used to decide whether to accept or reject the candidate. (3) Accept the candidate sample as the next state with probability $\xi$ by setting $x_{t+1} = \tilde{x}$. Otherwise reject the proposal and remain $x_{t+1} = x_t$.

Although MCMC makes it possible to sample from any desired distributions, its random-walk nature makes the Markov chain converge slowly to the stationary distribution $p(x)$.

In contrast, HMC employs physics-driven dynamics to explore the target distribution, which is much more efficient than the alternative MCMC methods. Before introducing HMC, we start out from an analogy of Hamiltonian systems in Neal et al. (2011) as follows. Suppose a hockey puck sliding over a surface of varying height and both the puck and the surface are frictionless. The state of the puck is determined by *potential energy* $U(\theta)$ and *kinetic energy* $\mathcal{K}(v)$, where $\theta$ and $v$ are the position and the momentum of the puck. The evolution equation is given by the Hamilton's equations:

$$\begin{cases} \frac{\partial \theta}{\partial t} = \frac{\partial H}{\partial v} = \nabla_v \mathcal{K}(v) \\ \frac{\partial v}{\partial t} = \frac{\partial H}{\partial \theta} = -\nabla_\theta U(\theta). \end{cases} \tag{3}$$

Due to the reversibility of Hamiltonian dynamics, the total energy of the system remains constant:

$$H(\theta, v) = U(\theta) + \mathcal{K}(v). \tag{4}$$

As for HMC, it contains three major parts: (1) Hamiltonian system construction; (2) Leapfrog integration; (3) Metropolis-Hastings correction. Firstly, the Hamiltonian is an energy function for the joint density of the variables of interest $\theta$ and auxiliary momentum variable $v$, so HMC defines a joint distribution via the concept of a canonical distribution:

$$p(\theta, v) \propto \exp\left(\frac{-H(\theta, v)}{\tau}\right), \tag{5}$$

where $\tau = 1$ for the common setting. Then, HMC discretizes the system and approximately simulates Eq. (3) over time via the leapfrog integrator. Finally, because of inaccuracies caused by the discretization, HMC performs Metropolis-Hastings Metropolis et al. (1953) correction without reducing the acceptance rate.

According to Eq. (4) and (5), the joint distribution can be divided into two parts:

$$p(\theta, v) \propto \exp\left(\frac{-U(\theta)}{\tau}\right) \exp\left(\frac{-\mathcal{K}(v)}{\tau}\right). \tag{6}$$

Since $\mathcal{K}(v)$ is an auxiliary term and always setting $\mathcal{K}(v) = v^T \mathbf{I}^{-1} v / 2$ with identity matrix $\mathbf{I}$ for standard HMC, our aim is that the potential energy $U(\theta)$ can be defined as $U(\theta) = -\log p(\theta)$ to explore the target density $p$ more efficiently than using a proposal probability distribution. If we can calculate $\nabla_\theta U(\theta) = -\frac{\partial \log(p(\theta))}{\partial \theta}$, then we can simulate Hamiltonian dynamics that can be used in an MCMC technique.

## 4.2 SIMULATING ADVERSARIAL EXAMPLES GENERATING BY HMC

Assume that the adversarial examples for $x$ with label $y$ are distributed over the solution space $\Omega$. Given any input pair $(x, y)$, for a specified model $f(\cdot) \in \mathcal{F}$ with fixed parameters, the adversary aims to find such examples $\tilde{x}$ that can mislead the model:

$$\Omega = \arg \max_{N(x) \subset \mathcal{N}(x)} \int J(\tilde{x}, y) \, p(\tilde{x}|x, y) \, d\tilde{x}, \tag{7}$$

where $\mathcal{N}(x)$ is the neighboring regions of $x$ and defined as $x' \in \mathcal{N}(x) := \left\{ \|x' - x\|_{1,2,\text{or} \infty} \leq \epsilon \right\}$. From the perspective of Bayesian statistics, we can make inference about adversarial examples over a solution space $\Omega$ from the posterior distribution of $\tilde{x}$ given the natural inputs $x$ and labels $y$.

$$\tilde{x} \sim p(\tilde{x}|x, y) \propto p(y|\tilde{x}) p(\tilde{x}|x), \quad \tilde{x} \in \Omega. \tag{8}$$

In Hamiltonian system, it becomes to generate samples from the joint distribution $p(\theta, v)$. Let $\theta = \tilde{x}$, according to Eq. (8) and (6), we can express the posterior distribution as a canonical distribution (with $\tau = 1$) using a potential energy function defined as:

$$\begin{aligned} U &= \frac{1}{N} \sum_{i=1}^{N} -\log p(y^{(i)}|\tilde{x}^{(i)}) - \log p(\tilde{x}|x) \\ &= J(\tilde{x}, y) - \log p(\tilde{x}|x). \end{aligned} \tag{9}$$

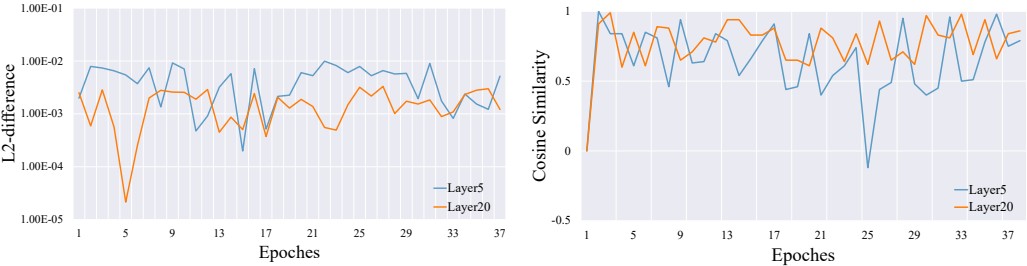

Figure 1: Measurement of TS (as defined in Definition 5) in different layers of ResNet34 on CIFAR10. For a layer, we measure the L2-difference and cosine similarity of the gradients running different lengths of trajectory.

Since $J(\tilde{x}, y)$ is the usual classification likelihood measure, the question remains how to define $p(\tilde{x}|x)$. A sensible choice is a uniform distribution over the $L_p$ ball around $x$, which means we can directly use a DNN classifier to construct a Hamiltonian system for adversarial examples generation as the base step of HMC.

Recall that the development of adversarial attacks is mainly based on the improvement of the vanilla fast gradient sign method, which derives I-FGSM, PGD and MI-FGSM. For clarity, we omit some details about the correction due to the constraint of adversarial examples. The core policy of the family of fast gradient sign methods is:

$$\tilde{x}_t = \tilde{x}_{t-1} + \varepsilon \cdot \text{sign}(g_t), \tag{10}$$

where $g_t$ is the gradient of $J$ at the $t$-th iteration, i.e., $\nabla_x J(\tilde{x}_{t-1}, y)$. It is clear that the above methods are the specialization of HMC by setting:

$$\begin{aligned} \theta_t = \tilde{x}_t, \quad v_t = g_t, \\ H(\theta, v) = J(\theta) + |v|. \end{aligned} \tag{11}$$

More specifically, I-FGSM can be considered as the degeneration of HMC, which explicitly updates the position item $\theta$ but implicitly changes the momentum item $v$ at every iteration. One of the derivation of I-FGSM, MI-FGSM, has explicitly updated both $\theta$ and $v$ by introducing $g_t = \mu g_{t-1} + \frac{1}{||\nabla J(\tilde{x}_{t-1}, y)||_1} \nabla J(\tilde{x}_{t-1}, y)$ after Eq. (10) at each step with the decay factor $\mu = 1$. The other derivative PGD runs Eq. (10) on a set of initial points $\tilde{x}_0 \in \left\{ \tilde{x}_0^{(1)}, \tilde{x}_0^{(2)}, \cdots, \tilde{x}_0^{(S)} \right\}$ adding different noises, which can be treated as a parallel HMC but the results are mutually independent.

## 5    ADVERSARIAL TRAINING USING CONTRASTIVE DIVERGENCE

As mentioned in Section 4, the inner maximization problem can be reformulated as the process of HMC. It is obvious that the high computational cost of adversarial training can be easily attributed to the long trajectory of MCMC searching for the stationary distribution of adversarial examples. Nevertheless, does training a robust model really need such a long trajectory?

To answer this question, we consider studying the gradient of the loss since the training procedure (obtaining $\nabla_v \mathcal{K}(v)$ and $\nabla_\theta U(\theta)$ and updating parameters of DNN) is a first-order method. To quantify the extent to which the parameters in a layer would change in reaction to the length of the trajectory, we measure the difference between the gradients of each layer running different lengths of trajectory. This leads to the following definition.

**Definition 5.1.** Let $W_1^{(K)}, ..., W_n^{(K)}$ be the parameters of each of the $n$ layers and $\left(\tilde{x}^{(k)}; y\right)$, $\left(\tilde{x}^{(k')}; y\right)$ be the batch of input-label pairs used to adversarially train the network. We define trajectory shift (TS) of activation $i$ along different lengths of trajectoriry $k$ and $k'$ to be the difference

$\left\| g_{k,i} - g'_{k',i} \right\|_{dist}$, where

$$g_{k,i} = \nabla_{W_i^{(k)}} J \left( W_1^{(k)}, \ldots, W_n^{(k)}; \tilde{x}^{(k)}, y \right)$$
$$g'_{k',i} = \nabla_{W_i^{(k')}} J \left( W_1^{(k')}, \ldots, W_n^{(k')}; \tilde{x}^{(k')}, y \right). \tag{12}$$

The difference between $g_{k,i}$ and $g'_{k,i}$ thus reflects the change in the optimization landscape of parameters $W_i$ caused by the changes to its input, which captures the shift of different lengths of trajectory that could have an influence on adversarial training. Equipped with this definition, we measure TS on ResNet34 trained with adversarial examples simulating by different lengths of trajectory ($k = 2, k' = 10$) throughout the training. Results are shown in Fig. 1. Although the situation in the bottom layer (e.g. layer5) is rather different than that in the top layer (e.g. layer20), both the direction and the magnitude of the gradients are quite close when simulating different lengths of trajectory. These evidences suggest that running a full trajectory for many steps is too inefficient since the model changes very slightly between parameter updates.

Thus, we might take advantage of that by initializing a HMC at the state in which it ended for the previous model. This initialization is often fairly close to the model distribution, even though the model has changed a bit in the parameter update. Besides, the high acceptance rate of HMC indicates that it is not necessary to run a long trajectory from the initial point. Therefore, we can simply run the chain for small (or even one) full step and then update the parameters to reduce the tendency of the chain to wander away from the initial distribution on the first step instead of running the full trajectory to equilibrium. We take small number $K$ of transitions from the data sample $\{x_i\}_i^n = 1$ as the initial values of the MCMC chains and then use these $K$-step MCMC samples to approximate the gradient for updating the parameters of the model. Algorithm1 summarizes the full algorithm.

Moreover, we also present a new training objective function $J_{cd}$, which minimizes the difference of KL divergence between two adjacent sampling steps to substitute the common KL loss:

$$J_{cd} = \rho(Q^0 \| Q^\infty) - \lambda(Q^1 \| Q^\infty), \tag{13}$$

where $\|$ denotes a Kullback-Leibler divergence and $\rho$ and $\lambda$ are the balanced factors. $Q_0$ and $Q_1$ are the output vector of DNN given input images $x + v_0$ and $x + v_K$. The intuitive motivation for using this $J_{cd}$ is that we would like every state in HMC exploring to leave the initial distribution $Q_0$ and $Q^0 \| Q^\infty$ would never exceed $Q^1 \| Q^\infty$ until $Q_1$ achieves the equilibrium distribution. We set $\lambda = 2, \rho = 1$ and analyze how this objective function influences the partial derivative of the output probability vector with respect to the input. Due to the fact that the equilibrium distribution $Q^\infty$ is considered as a fixed distribution and the chain rule, we only need to focus on the derivative of the softmax output vector with respect to its input vector in the last layer as follows:

$$\begin{aligned} \nabla U_{\text{last}} &= 2 \sum_c y_c \frac{\partial \log f_\omega(\tilde{x}^K)_c}{\partial \tilde{x}'} - \sum_c y_c \frac{\partial \log f_{\tilde{\omega}}(\tilde{x})_c}{\partial \tilde{x}'} \\ &= 2 f_\omega(\tilde{x}^K)_c \sum_c y_c - f_{\tilde{\omega}}(\tilde{x})_c \sum_c y_c - y \\ &= f_\omega(x^K) - (y - \Delta f), \end{aligned} \tag{14}$$

where $\Delta f = f_\omega(x^K) - f_{\tilde{\omega}}(\tilde{x})$. Based on this abbreviation, we can easily get the relationship between Eq. (14) and $\frac{\partial J_{ce}}{\partial \tilde{x}'} = f_\omega(x^K) - y$. For each adversarial example generation, Eq. (14) makes an amendment of $y$ which is determined by the difference of current and the last $K$-step HMC samples output probability. Since $f_\omega$ and $f_\omega(x)$ are more closer to $f^*$ and $y^*$ than $f_{\tilde{\omega}}$ and $f_{\tilde{\omega}}(x)$, each update of $\tilde{x}$ would be better corrected.

## 6 EXPERIMENTAL RESULTS

In this section, we focus on the ImageNet Deng et al. (2009), CIFAR10 Krizhevsky & Hinton (2009) and MNIST LeCun (1998) datasets with extensive experiments to validate the effectiveness of the proposed methods. For most part of experiments, we compare three standard adversarial training baselinesMadry et al. (2018); Zhang et al. (2019d); Rice et al. (2020) and three advanced acceleration methodsShafahi et al. (2019); Zhang et al. (2019b); Wong et al. (2020) with our ATCD. More details about experiment setup can refer to Appendix A.1. Extensive ablation studies on CIFAR10 can also be found in Appendix A.3.

---

**Algorithm 1** Adversarial Training using Contrastive Divergence (ATCD)

---

**Input:** A DNN classifier $f_\omega(\cdot)$ with initial learnable parameters $\omega_0$; training data $x$ with visible label $y$; number of epochs $N$; length of trajectory $K$; repeat time $T$; magnitude of perturbation $\varepsilon$; learning rate $\kappa$; step size $\alpha$.
*/\*Stage-0: Construct Hamiltonian system\*/*
$U(\theta, \omega, \tilde{\omega}, y, k) = -J_{cd}\left(f_\omega(\theta^{k-1}), f_{\tilde{\omega}}(\theta^K), y\right), \mathcal{K}(v) = |v|$
Initialize $\omega = \tilde{\omega} = \omega_0, \theta^K = \theta^0$.
**for** epoch$= 1 \cdots N/(TK)$ **do**
    $\theta^0 \leftarrow x + v_0, v_0 \sim \text{Uniform}(-\varepsilon, \varepsilon)$.
    **for** $t = 1$ **to** $T$ **do**
        */\*Stage-1: Generate adversarial examples by K-step contrastive divergence\*/*
        **for** $k = 1$ **to** $K$ **do**
            $\theta^k \leftarrow \theta^{k-1} + \varepsilon \cdot \nabla \mathcal{K}(v_{t-1})$
            $v_t \leftarrow v_{t-1} - \alpha \nabla U(\theta, \omega, \tilde{\omega}, y, k)$
            $v_t \leftarrow \text{clip}(v_t, -\varepsilon, \varepsilon)$
        **end for**
        $\left(\theta^K, v_t\right) = \left(\theta^k, v_t\right)$, M-H step decides whether it should be accepted or rejected.
        */\*Stage-2: Update parameters of DNN by generated adversarial examples\*/*
        $\boldsymbol{g}_\omega \leftarrow \mathbb{E}_{(\theta, y)}\left[\nabla_\omega J_{ce}(f_\omega(\theta^K), y)\right]$
        $\tilde{\omega} \leftarrow \omega$
        $\omega \leftarrow \omega - \kappa \boldsymbol{g}_\omega$
    **end for**
**end for**

---

## 6.1 IMAGENET

For ImageNet, we fix the total loop times $T * K = 4$ same as Free-4 Shafahi et al. (2019) for fair comparison. We report average over the final 3 evaluation. Comparison between free adversarial training and ours are shown in Table 1. Although the 2-PGD trained ResNet-50 model still maintains its leading role in the best robust accuracy, it takes three times longer than our ATCD method. Actually, when compared with its high computational cost of ImageNet training, this performance gain can be considered inefficient or even impractical for resource limited entities. Besides, ATCD is an anytime algorithm expected to find better and better solutions the longer it keeps running. We also compare ResNet-50 model trained by our ATCD method with the Free-4 trained, model trained by ATCD produces much more robust models than Free-4 against different attacks in almost the same order of time. Though Fast-FGSM achieves a sterling acceleration, both its clean accuracy and robust accuracy are not satisfactory enough.

| Methods | Clean Data | PGD-10 | PGD-20 | PGD-50 | MI-FGSM-20 | Speed (mins) |
|---|---|---|---|---|---|---|
| Natural train | 75.34% | 0.14% | 0.06% | 0.03% | 0.03% | 1437 |
| PGD Madry et al. (2018) | **63.95%** | **36.89%** | **36.44%** | **36.17%** | **35.29%** | 8928 |
| Free-4 Shafahi et al. (2019) | **60.26%** | 31.12% | 30.29% | 30.07% | 29.43% | 2745 |
| Fast-FGSM Wong et al. (2020) with apex | 55.68% | 30.23% | 29.07% | 28.91.% | 27.88% | **718** |
| ATCD-2-1 with/without apex | 59.23% | **35.91%** | **35.72%** | **35.76%** | **34.67%** | **1229** / 2992 |

Table 1: Validation accuracy and robustness of ResNet50 on ImageNet. We report average over the final 3 runs. The maximum perturbation of all the attackers is $\varepsilon = 4/255$. The best results are in red while the second best results are in blue. Our ATCD achieves a trade-off between efficiency and accuracy.

## 6.2 CIFAR10

For CIFAR10, we fix the total loop times $T * K = 8$ same as Free-8 Shafahi et al. (2019) for fair comparison and show the training time of all methods. We calculate the deviation value of final 5 evaluation and report average over 5 runs with different restarts. Results on Wide ResNet34 Zagoruyko & Komodakis (2016) are summarized in Table 2.

From the table, we can see that the naturally trained model (without any adversarial examples) is vulnerable to all the attacks, while the baseline adversarial training methods (PGD, TRADES, Robust-Overfitting) produce robust models that are effective to defend PGD attacks and goodish to other type of attacks. Three advanced acceleration methods (Free, YOPO/TRADES+YOPO, Fast-FGSM) and ours can be at least 4∼5 times faster than previous adversarial training methods. Although

| Methods | Clean Data | PGD-20 | MI-FGSM-20 | CW | AA | RayS | Speed (mins) |
|---|---|---|---|---|---|---|---|
| Natural train | 94.58% | 0.00% | 0.00% | 0.00% | 0.00% | 0.00% | 212 |
| PGD-10 Madry et al. (2018) | **87.11%±0.37%** | 48.4%±0.22% | 44.37%±0.11% | 45.91%±0.14% | 43.88%±0.15% | 49.91%±0.37% | 2602 |
| TRADES-10 Zhang et al. (2019d) | 85.63%±0.44% | 53.21%±0.57% | 52.22%±0.27% | 52.08%±0.39% | **52.67%±0.27%** | 56.91%±0.23% | 2695 |
| Robust-Overfitting Rice et al. (2020) | 85.21%±0.66% | **57.46%±0.71%** | **54.38%±0.52%** | **54.81%±0.51%** | **53.14%±0.31%** | **58.13%±0.37%** | 4500 |
| Free-8 Shafahi et al. (2019) | 84.29%±1.44% | 47.8%±1.32% | 47.01%±0.19% | 46.71%±0.22% | 42.53%±0.37% | 51.37%±0.38% | 646 |
| YOPO-5-3 Zhang et al. (2019a) | 84.72%±1.23% | 46.4%±1.49% | 47.24%±0.25% | 47.5%±0.37% | 44.44%±0.29% | 50.81%±1.55% | 457 |
| TRADES+YOPO-3-4 Zhang et al. (2019a) | 87.55%±∞% | 48.86%±∞% | 48.13%±∞% | 49.53%±∞% | 49.53%±∞% | 51.19%±∞% | 1231 |
| Fast-FGSM Wong et al. (2020) with apex | 83.21%±0.66% | 46.46%±0.71% | 45.38%±0.52% | 45.81%±0.51% | 43.01%±0.17% | 48.78%±0.51% | **23** |
| ATCD-4-1 with/without apex | **86.09%±0.27%** | **54.2%±0.44%** | **52.73%±0.21%** | **52.62%±0.15%** | 50.2%±0.11% | **57.11%±0.24%** | **167** / 672 |

Table 2: Validation accuracy and robustness of Wide ResNet34 on CIFAR10. The maximum perturbation of all the attackers is $\varepsilon = 8/255$. We report average over 5 runs with different restarts. The "∞" error bars mean the method sometimes cannot converge during the training process. The best results are in red while the second best results are in blue.

Fast-FGSM achieves the best speed improvement (using the apex library), our ATCD is the only method that even greatly boost the robust accuracy in a reasonable training speed. Similar success also appear in different architectures (see in Appendix A.3.1). We also perform the evaluation among different methods on both clean accuracy (i.e. accuracy on natural images) and robust accuracy (i.e. accuracy on adversarial examples) after every training epoch and show the remarkable reliability about our ATCD in Fig. 2. We further emphasize that although YOPO may be computationally cheaper when compared to conventional approaches and other methods, it is clear that the curve of YOPO vibrates greatly and frequently, which implies the training scheme of YOPO should be carefully designed to achieve stable results. This is also reflected in Table 2 when compared with the error bars of YOPO (TRADES+YOPO) and ours.

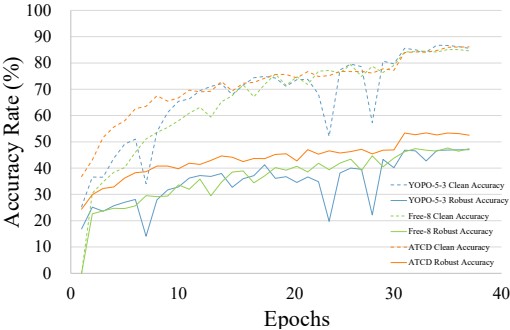

Figure 2: Comparison with different advanced fast adversarial training methods. We use PGD-20 as the attacker and report their clean accuracy and robust accuracy. Solid lines represent the robust accuracy and dashed lines represent the clean accuracy.

## 6.3 MNIST

We also investigate our ATCD method on MNIST. PGD-40 still has comparable clean accuracy and robust accuracy among all the methods, but its computational cost is significantly higher than other training methods. Our method still achieves a good trade-off between efficiency and robust accuracy.

| | Clean Data | PGD-40 | CW | Speed (secs) |
|---|---|---|---|---|
| Natural train | 99.98% | 0.00% | 0.00% | 196 |
| PGD-40 Madry et al. (2018) | **99.50%** | **97.17%** | **93.27%** | 1877 |
| Free-10 Shafahi et al. (2019) | 98.29% | 95.33% | 92.66% | **415** |
| YOFO-5-10 Zhang et al. (2019a) | **99.98%** | 94.79% | 92.58% | **312** |
| ATCD-2-1 | 99.36% | **97.48%** | **94.77%** | 441 |

Table 3: Validation accuracy and robustness of a small CNN on MNIST. The maximum perturbation of all the attackers is $\varepsilon = 0.3$. The best results are in red while the second best results are in blue.

## 7 CONCLUSION

In this paper, we reformulate the generation of adversarial examples as a MCMC process and present a new adversarial learning method called ATCD, which approaches equilibrium distribution of adversarial examples with only few iterations by building from small modifications of the standard Contrastive Divergence. Extensive results with comparisons on various datasets show that ATCD achieves a trade-off between efficiency and accuracy in adversarial training.

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

## A  APPENDIX

### A.1  EXPERIMENT SETUP

All the experiments are taken on a single NVIDIA GeForce GTX Xp GPU. Results of Free training Shafahi et al. (2019), TRADES-10 Zhang et al. (2019d), YOPO Zhang et al. (2019b), Fast-FGSM Wong et al. (2020) and Robust-Overfitting Rice et al. (2020) are reproduced on our local machines

but completely follow their official codes[1][2][3][4][5]. Note that in our experiments, we do *not* use any label-smoothing or other tricks to boost the robustness accuracy since we would like to fairly compare PGD, Free and YOPO with our ATCD method. These extra tricks and adversarial interpolation schemes can be added to improve results for both approaches but *not* the focus of our paper.

**ImageNet setup.** The ILSVRC 2012 classification dataset contains ~1.28M training images and 50k validation images labeled with 1,000 classes. Following the common practice, we perform horizontal flip, scale, and aspect ratio augmentation for training images and apply a single center crop of $224 \times 224$ pixels during evaluation. We choose the standard Resnet50 as the target model. For all methods, we use a batch size of 256, and SGD optimizer with momentum 0.9 and a weight decay of 1e-4. The initial learning rate is 0.1 and the learning rate is decayed by 10 every $30/TK$ epochs. We also set step size $\epsilon = 4/255$ and magnitude of perturbation $\varepsilon = 4/255$ based on $L_\infty$ norm. We compare to the best performing configuration of Free adversarial training which uses 4 minibatch replays over 96 epochs of training. That means we only run $N/TK = 24$ passes over the whole dataset for both Free training and ATCD.

**CIFAR10 setup.** The CIFAR10 dataset is a widely used dataset consisting of 60,000 colour images of 10 categories. Each category has 6,000 images. We choose the standard Wide ResNet-34 and Preact-ResNet18 following previous works Madry et al. (2018); Zhang et al. (2019a). We use a similar scheme that was used in YOPO Zhang et al. (2019b): we train models for 40 epochs and the initial learning rate is set to 0.2, reduced by 10 times at epoch 30 and 36. For PGD adversarial training, we set the total epoch number $N = 105$ and 10 steps of PGD with step size $\epsilon = 2/255$ and magnitude of perturbation $\varepsilon = 8/255$ as a common practice. The initial learning rate is set to 5e-2, reduced by 10 times at epoch 79, 90 and 100. We use a batch size of 256, a weight decay of 5e-4 and a momentum of 0.9 for both algorithm. For evaluating, we test our model's robustness under CW Carlini & Wagner (2017), MI-FGSM Dong et al. (2018), AutoAttck (AA)cro (2020), RaySChen & Gu (2020) and 20 steps of PGD with step size $\epsilon = 2/255$ and magnitude of perturbation $\varepsilon = 8/255$ based on $L_\infty$ norm.

**MNIST setup.** MNIST is a database for handwritten digit classification. It consists of 60,000 training images and 10,000 test images, which are all $28 \times 28$ greyscale images, representing the digits 0-9. We choose a simple ConvNet with four convolutional layers followed by three fully connected layers, which is same as Zhang et al. (2019a). For PGD adversarial training, we train the models for 55 epochs. The initial learning rate is set to 0.1, reduced by 10 times at epoch 45. We use a batch size of 256, a weight decay of 5e-4 and a momentum of 0.9. For evaluating, we perform a PGD-40 and CW attack against our model and set the size of perturbation as $\varepsilon = 0.3$ based on $L_\infty$ norm as a common practice Madry et al. (2018); Zhang et al. (2019a;c).

## A.2 DETAILS ABOUT TRAJECTORY SHIFT

Trajectory Shift (TS) is our proposed metric to measure the difference of running various lengths of trajectory. To reflect the change in the optimization landscape of parameters of a network during the course of training, we essentially take different lengths in MCMC simulating and measure the gradient obtained by each backward.

Specifically, suppose we would like to investigate the TS of the i-th layer and the lengths of different trajectories are $K$ and $K'$, what we need to do is to generate adversarial inputs according to the $K$ and $K'$-step trajectory of MCMC simulating. When the trajectory simulation is finished, we perform gradient backward based on the potential energy function and adversarial examples at current state, and then gather all the gradients in the mini-batch. Finally, all gradients will be added and averaged throughout the dataset when both trajectory simulating and parameters updating are completed. So the trajectory shift is the difference of these two averaged results. The whole process can be summarized

---

[1] Code from https://github.com/ashafahi/free_adv_train.

[2] Code from https://github.com/a1600012888/YOPO-You-Only-Propagate-Once.

[3] Code from https://github.com/yaodongyu/TRADES

[4] Code from https://github.com/locuslab/fast_adversarial

[5] Code from https://github.com/locuslab/robust_overfitting

as the following formualtion:

$$TS = \frac{1}{|\mathcal{D}|} \sum_{b=1}^{\frac{|\mathcal{D}|}{B}} || \frac{1}{T} \sum_{k=1}^{T} \nabla_{W_i^{(k)}} J\left(W_1^{(k)}, \ldots, W_n^{(k)}; \tilde{x}^{(k)}, y\right) - $$

$$\frac{1}{\lfloor \frac{TK}{K'} \rfloor} \sum_{k'=1}^{\lfloor \frac{TK}{K'} \rfloor} \nabla_{W_i^{(k')}} J\left(W_1^{(k')}, \ldots, W_n^{(k')}; \tilde{x}^{(k')}, y\right) ||_{dist}, \tag{15}$$

where $B$ is the size of mini-batch. Note that in order to eliminate some unnecessary effects, we use the same random seed and share initialization parameters for two sets of trajectory with different lengths, and the input samples of each mini-batch are exactly the same.

## A.3 ABLATION STUDIES

### A.3.1 DIFFERENT ARCHITECTURES

Besides the success on Wide ResNet34 Zagoruyko & Komodakis (2016), we also achieves similar acceleration against PGD-10 in Preact-ResNet18 He et al. (2016a), as shown in Table 4. ATCD can achieve more aggressive robust accuracy boost with only a slight drop in clean accuracy when compared with Free and YOPO. The word "natural train" in the table means the models are trained by using only natural (clean) images as the tradition classification task always does.

| Methods | Clean Data | PGD-20 | MI-FGSM-20 | CW | Speed (mins) |
|---|---|---|---|---|---|
| Natural train | 93.78% | 0.00% | 0.00% | 0.00% | 47 |
| PGD-10 Madry et al. (2018) | 84.96%±0.12% | 41.58%±0.11% | 39.47%±0.27% | 58.88%±0.33% | 132 |
| TRADES-10 Zhang et al. (2019d) | **85.05%±0.48%** | 43.57%±0.62% | 42.33%±0.28% | 57.21%±0.35% | 607 |
| Robust-Overfitting Rice et al. (2020) | **85.15%±1.03%** | **46.73%±0.89%** | **45.38%±0.52%** | **60.81%±0.51%** | 1033 |
| Free-8 Shafahi et al. (2019) | 82.44%±0.37% | 42.07%±0.44% | 41.88%±0.53% | 57.02%±0.22% | 110 |
| YOPO-5-3 Zhang et al. (2019a) | 82.65%±0.75% | 42.56%±0.83% | 41.85%±0.44% | 56.93%±0.71% | 66 |
| TRADES+YOPO-3-4 Zhang et al. (2019a) | 84.73%±∞% | 43.56%±∞% | 42.13%±∞% | 56.93%±∞% | 231 |
| Fast-FGSM Wong et al. (2020) with apex | 80.91%±0.77% | 45.74%±0.63% | 43.27%±0.51% | 56.57%±0.28% | **8** |
| ATCD-2-1 with/without apex | 81.54%±0.31% | **49.37%±0.27%** | **48.56%±0.09%** | **61.28%±0.29%** | **28** / 114 |

Table 4: Validation accuracy and robustness of Preact-ResNet18 on CIFAR10. The maximum perturbation of all the attackers is $\varepsilon = 8/255$. We report average over 5 runs with different restarts. The best results are in red while the second best results are in blue.

### A.3.2 INFLUENCE OF UPDATING FREQUENCY AND TRAJECTORY LENGTH

We study the effect of updating iteration $T$ and the length of trajectory $K$ on the accuracy rate for our ATCD. We fix $T * K = 8$ and apply several configurations. Results are shown in Table 5 and Figure. 3. It can be observed that (1) The best accuracy rate of ATCD on both clean images and PGD adversarial examples appears at $T = 4, K = 2$. This may result from the balance it achieves between updating frequency and the length of trajectory when $T * K$ is limited. (2) We would like to point out Free training is a special case of ATCD, except that ATCD has a new objective function for approaching equilibrium distribution and can freely control the updating frequency and the length of MCMC trajectory. We compare our ATCD under several configurations with Free-8 (at the last row, and that's the reason why we fix $T * K = 8$). Note that our ATCD method surpass Free-8 in all cases, which shows that it is necessary to seperate parameters updating and adversarial examples generating instead of treating the service as one like Free model and $J_{cd}$ is helpful for the robustness.

### A.3.3 INFLUENCE OF M-H RESAMPLING

The proposed method has a new adversarial example generation objective and the M-H resampling. In order to find out what actually causes the performance improvement, we ablate the behavior of the generation objective and the M-H resampling. This ablation is in Table 6. We investigate the above two components for training Wide ResNet34 on CIFAR10. When adding the CD generation objective in the procedure of training, the accuracy increases greatly — e.g., increasing from 47.8% to 53.11% under 20-iteration PGD attacks. On the other hand, adding the M-H resampling strategy makes slight

| (T,K) | Clean Data | PGD-20 |
|---|---|---|
| (1,8) | 84.68%±0.57% | 48.8%±0.72% |
| (2,4) | 85.09%±0.27% | 51.44%±0.48% |
| (4,2) | **86.39%±0.27%** | **54.2%±0.44%** |
| (8,1) | 84.88%±0.25% | 49.29%±0.43% |
| (8,1) w/o $J_{cd}$ | 84.29%±1.44% | 47.8%±1.32% |

Table 5: Validation accuracy and robustness of Wide ResNet34 on CIFAR10. We report average over 5 runs. The best result under different attack methods is in bold.

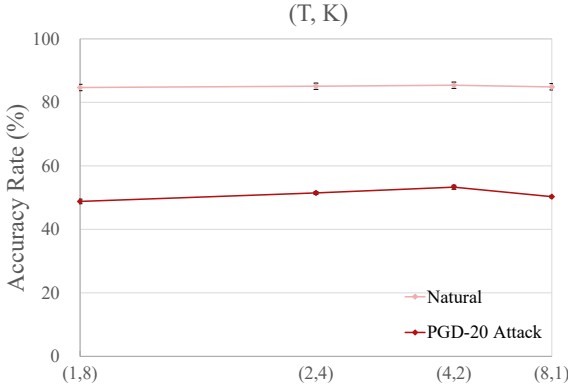

Figure 3: The accuracy rates of Wide ResNet34 trained by ATCD with different configurations $(T, K)$ on CIFAR10.

improvement on the robust accuracy, and it has a considerable increase on the clean accuracy. These results suggest that both of them bring benefit for the clean and the robust accuracy. As samples in our proposed method stay in (and return large numbers of samples from) high-density regions of the candidate distribution while only occasionally visiting low-density regions by the acceptance probability, which can can decipt the underlying distribution of adversarial examples. It appears essential to properly combine the CD loss with the M-H resampling in the training procedure.

| Method | Clean Data | PGD-20 |
|---|---|---|
| Baseline | 84.29% | 47.8% |
| + CD loss | 85.05% | 53.11% |
| + MH resampling | 86.39% | 54.2% |

Table 6: Effectiveness verification of the CD loss and the M-H resampling on CIFAR10. We use Wide ResNet34 as the target model and the baseline method is Free-8

### A.3.4 DIFFERENT NUMBERS OF ATTACK ITERATIONS

To analyze the influence of the number of attack iterations on different adversarially trained models, we range the number of iterations of PGD attacker from 10 to 200. Figure 4 shows the robust accuracy of different adversarially trained models. We fix $\varepsilon = 8/255$ and $\varepsilon = 2/255$, and change the iteration number ranging in $\{10, 20, 50, 100, 200\}$. From the plots, we see that no matter how larger the attack iteration is, the checkpoint which achieves the high robust accuracy under one certain configuration still has higher accuracy than other models in different iteration configurations.

### A.3.5 DIFFERENT VALUES OF EPSILON

Figure 5 shows the robust accuracy of models using various state-of-the-art adversarial training methods. We performed thorough experiments over epsilon parameters $\varepsilon = \{8/255, 16/255, 32/255, 64/255, 128/255\}$, and found that the robust accuracy drop dramatically when $\varepsilon \geq 32/255$ especially all of the adversarial training acceleration methods. In spite of this,

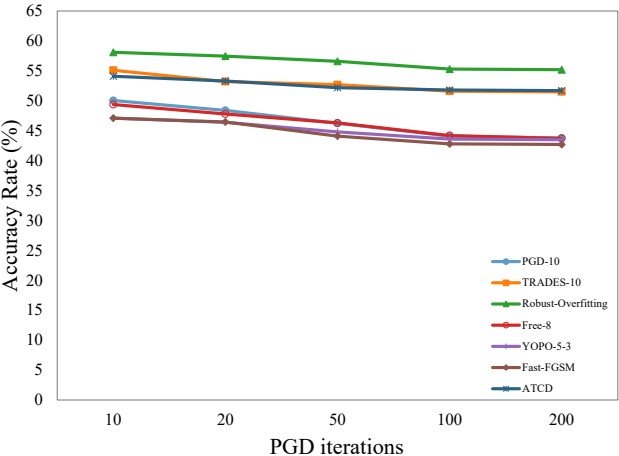

Figure 4: The accuracy rates of Wide ResNet34 trained by different adversarial training methods with different epsilon on CIFAR10. We use PGD with $\varepsilon = 8/255$ and $\varepsilon = 2/255$ to performe thorough experiments over the number of attack iterations $\varepsilon = \{10, 20, 50, 100, 200\}$.

our ATCD method still aces any other acceleration method, even outperforming the standard PGD training.

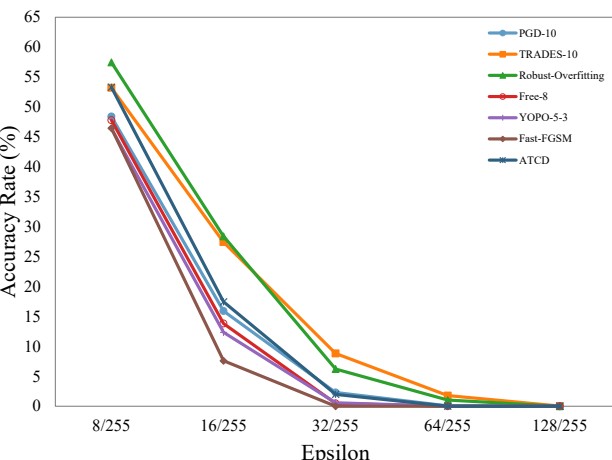

Figure 5: The accuracy rates of Wide ResNet34 trained by different adversarial training methods with different epsilon on CIFAR10. We use PGD with 100 iterations and well-tuned steps as attacker to perform thorough experiments over epsilon parameters $\varepsilon = \{8/255, 16/255, 32/255, 64/255, 128/255\}$.

### A.3.6 SENSITIVE ANALYSIS OF $\lambda$ AND $\rho$

In the ATCD algorithm, the main parameters include the balance factors $\lambda$ and $\rho$. Figure 6 plots the results of the parameter sensitivity analysis. We broadly investigate different ratios of $\lambda/\rho$ ranging from 1/32 to 32. It can be found that when the parameters are changed, the algorithm performs stably. Models can achieve rather better accuracy when the ratio of $\lambda/\rho$ approaches 1 and the best performance appears when $\lambda/\rho = 4$. It is worth mentioning that our proposed objective function will degenerate into the original generation objective widely used in adversarial examples generation stage when $\rho \ll \lambda$. Therefore, our ATCD will be practically equal to Free with resampling when $\lambda/\rho = 32$, which outperforms Free itself and shows the benefits brought from the Metropolis-Hastings correction.

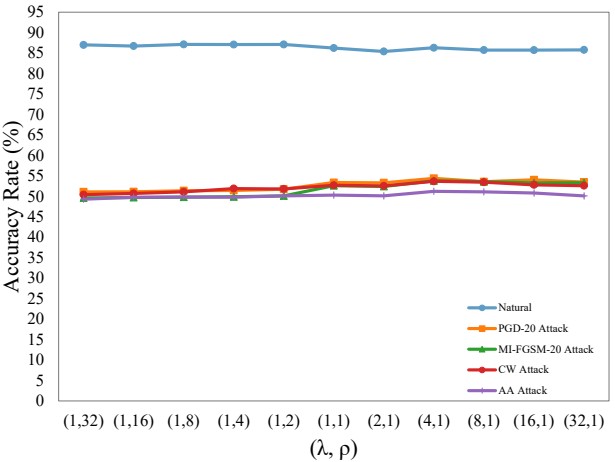

Figure 6: The accuracy rates of Wide ResNet34 trained by our method with different ratios of $\lambda/\rho$ ranging from 1/32 to 32 on CIFAR10. We use several attackers including PGD-20, MI-FGSM-20, CW and AA to validate the stability of parameters.

## A.4 COMPLEXITY ANALYSIS

A typical adversarial training method contains three major steps: forward propagation, backpropagation, and adversarial examples generation. Suppose that a deep neural network contains $L$ layers (including input and output layer). $d_l$ denotes the number of nodes of in each layer, where $l \in \{1, \cdots, L\}$. Assume the dataset has $Z$ training examples and the number of training epochs is $N$. Here we ignore the run-time of batch normalization for simplification.

**Forward propagation.** We see that for each layer a matrix multiplication, and an activation function is computed. It is easy to know that naive matrix multiplication has a asymptotic run-time of $O\left(d^3\right)$, and the activation operator $g$ is an elementwise function, we know that it has a run-time of $O\left(d\right)$. So the total forward run-time therefore becomes $O_{\text{forward}} = NZ * (O_{\text{mul}} + O_{\text{activation}}) = NZ * (\sum_{l=2}^{L} \left(d^{(l)}d^{(l-1)}d^{(l-2)}\right) + \left(d^{(1)}d^{(0)}1\right) + \sum_{l=1}^{L} \left(d^{(l)}\right)) = NZL(d^3 + d)$.

**Backpropagation.** The run-time complexity of backpropagation can be computed in a similar manner. Suppose $z_i^{(l)}$ is the raw output signal of the $i$-th neuron in layer $l$ before the activation function has been applied. We introduce a new variable $\delta_i^{(l)}$ which is the error-sum of neuron $i$ in layer $l$. So the total backpropagation run-time can be calculated as $O_{\text{backward}} = NZ * (O_{\text{middle\_layers}} + O_{\text{last\_layer}} + O_{\text{update}}) = NZ[(L-1) * O\left(\left(\omega^{(l+1)^T}\delta^{(l+1)}\right) \odot g'\left(z^{(l)}\right)\right) + O\left(\nabla_f^{(L)} \odot g'\left(z^{(L)}\right)\right) + \sum_{l=1}^{L} \left(d^{(l)}\right)] = NZL(\frac{L-1}{L}d^3 + \frac{1}{L}d^2 + d)$.

**Adversarial Training.** Recall that adversarial training is used adversarial examples to train DNN models to improve the robustness. So the major difference between adversarial training and normal training is that adversarial training contains the process of adversarial examples generation. Here we show several methods of adversarial examples generation to compare their time complexity.

- **FGSM**: Since FGSM finds the adversarial perturbation in the direction of the loss gradient only once, FGSM includes two full forward propagation, one backpropagation without weight updating, and one backpropagation with weight updating. So the total run-time of FGSM is: $O_{\text{FGSM}} = 2NZL[(d^3 + d) + \frac{L-1}{L}d^3 + \frac{1}{L}d^2 + \frac{1}{2}d]$.

- **PGD**: PGD can be considered as an iterative version of FGSM repeating $r$ times updating procedure of adversarial examples generation with several random restarts. That means PGD includes one full forward propagation, one backpropagation with weight updating, $r$ times forward propagation and backpropagation without weight updating. Thus, the total run-time of PGD can be computed by: $O_{\text{PGD}} = NZL(r+1)[(d^3 + d) + \frac{L-1}{L}d^3 + \frac{1}{L}d^2 + \frac{1}{r+1}d]$.

- **YOPO**: The major time cost of PGD results from conducting $T$ sweeps of forward and backward propagation. YOPO restricts most of the forward and back propagation within the first layer of DNN during adversary updates, which effectively reduces the total number of full forward and backward propagation. The total number of full forward propagation is only $m$ apart from $n$ times forward propagation at the first layer. The backward process is divided into $m$ times backpropagation except the first layer and $n$ times backpropagation at the first layer. Therefore, the total run-time of YOPO can be formulated as: $O_{\text{YOPO}} = NZ(m+1)[L(d^3+d) + (L-1)(\frac{L-1}{L}d^3 + \frac{1}{L}d^2 + d) + n(d^3 + d + d^2)]$.

- **ATCD**: Our ATCD method relies on the replay of current samples and the searching trajectory of HMC. So the total run-time of ATCD can be easily expressed by: $O_{\text{ATCD}} = N'ZLK(T+1)[(d^3+d) + \frac{L-1}{L}d^3 + \frac{1}{L}d^2 + \frac{1}{K(T+1)}d]$. Because of $N' = \frac{N}{TK}$, our ATCD is $\frac{r+1}{K(T+1)}$ times faster than PGD but $\frac{K(T+1)}{2}$ slower than FGSM.

### A.5    Long or Short-run Trajectory?

Both long and short-run trajectory of HMC simulating have been able to fit with contrastive adversarial training. In our experiments we use $K = 2$ to achieve the trade-off between performance and efficiency. But we have found that if condition allows (without computation or time being limited) and just pursue the best performance, long-run HMC with Metropolis-Hastings correction are preferable in terms of equilibrium distribution.

As shown in Figure 7, we fix the number of parameters updating as $T = 4$ and sweep over the length of trajectory $K$ from 1 to 8 (almost in line with PGD adversarial training). In all configurations, ATCD has much higher robust accuracy than Free. This shows evidence that ATCD is a more effective adversarial training method to improve the robustness of models compared to Free. We also find that by using contrastive divergence, we could reduce the lengths of MCMC trajectory per iteration when compared to PGD-10 and still allow for relatively high robustness. This gives a 3x speedup over our fastest short-run ATCD ($K = 1$). Further, robust accuracies against different adversarial attacks share the similar trend and ATCD using longest trajectory has the best result. So if time allows, we recommend using a long-run trajectory of HMC with Metropolis-Hastings correction to better approach equilibrium distribution. But even so, the clean accuracy of adversarially trained model still decreases with the growth of the length of simulating trajectory. Besides, some generated adversarial examples will be recalled by Metropolis-Hastings correction when the length of trajectory becomes longer, which would increase the additional time cost. Overall, the short-run trajectory is slightly less robust than the long-run trajectory, but in total, the short-run trajectory increases in speed and maintains the suitable clean accuracy to be a worthwhile trade-off.

Another interesting observation we found while using long-run trajectory of HMC is that the model would use the length of the trajectory to perform like "clustering". We discovered that when generating adversarial examples (the inner loop in Algorithm 2) on CIFAR10, these samples are always identified as one specific category (e.g. trunk) at very beginning of trajectory. When the length of trajectory grows, they could almost always be identified as another category (e.g. deer). However, if we use short-run trajectory, this phenomenon will not happen. This behavior is likely some compromise of ATCD since contrastive adversarial training with short-run trajectory accelerates the process of equilibrium distribution by discarding the solution space exploring and Metropolis-Hastings correction.

### A.6    What is the Advantage of HMC Framework for Adversarial Learning?

In this section we describe the advantage of reformulating adversarial examples generation from the perspective of HMC. In addition to transforming the process of adversarial training into the trajectory of MCMC simulating, thereby introducing more well studied solutions of high computational cost of MCMC, HMC also brings the following advantages:

First, we reinterpret the adversarial examples generation as the solution space searching through the conservation transformation of two kinds of energy. If we further consider a standard discriminative classifier of $p(y|x)$ as an energy based model for the joint distribution $p(x, y)$, we can apply our framework in both discriminative and generative modeling. That means we enable this framework

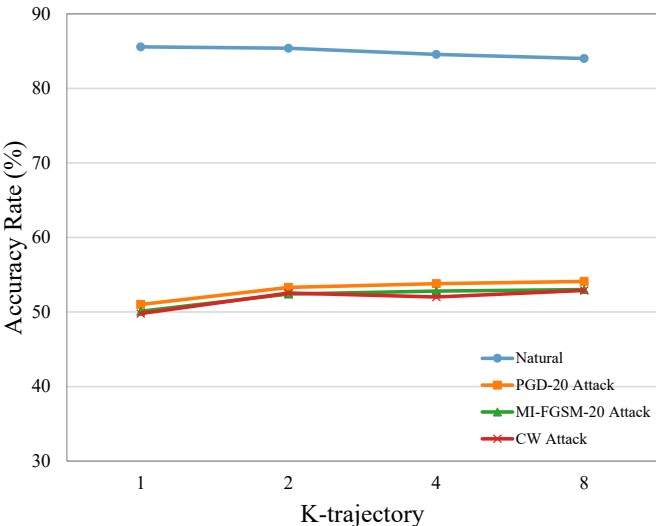

Figure 7: The accuracy rates of Wide ResNet34 trained by ATCD with different lengths of trajectory on CIFAR10. Basically, we could find that the robust accuracy of model increases with the length of the trajectory.

not only to improve the robustness of modern classifier, but also to generate samples as generative models.

Second, when simulating adversarial examples generation in HMC framework, different types of adversarial attacks can be considered as special cases of HMC. Actually, there will be a more powerful searching algorithm if a new potential or kinetic (or both) energy is designed Bhagoji et al. (2017); Carlini & Wagner (2017); Zheng et al. (2019); Rony et al. (2019), e.g. mapping the clipped gradient descent into tanh space or adding KL-divergence term.

Third, the unique advantage brought by HMC itself is that it can better explore the distribution of adversarial examples, which is different from heuristic algorithm (e.g. PGD with random restarts). That means we can design a more powerful adversarial attack from the view of HMC. To further verify this point, we apply a generating method for adversarial examples based on HMC to attack the real-world celebrity recognition APIs in Clarifai[6]. These celebrity recognition APIs allow users to upload any face images and recognize the identity of them with confidence score. The users have no knowledge about the dataset and types of models used behind these online systems. We choose 10 pairs of images from the LFW dataset and learn perturbations from local facenet model to launch targeted attack, whose goal is to mislead the API to recognize the adversarial images as our selected identity. We randomly pick up 10 celebrities as victims from Google and 10 existing celebrities as targets from LFW, ensuring that all colors and genders are taken into account. Then we apply the same strategy as Geekpwn CAAD 2018 method that pulls victims towards their corresponding targets by the inner product of their feature vectors and generates noise to them. Finally, we examine their categories and confidence scores by uploading these adversarial examples to the online systems API. We fix $\varepsilon = 16$ and total iteration number $N = 100$. It is worth mentioning that we generate a sequence of adversarial examples to show how well HMC-based method explores the space of adversarial examples. Results are shown in Fig. 8, 9 and 10.

Figure 8 presents three celebrity pairs where both Geekpwn CAAD 2018 method and our HMC-based method successfully fool the face recognition system to recognize them as the target celebrities. But the confidence score of each pair generated by Geekpwn CAAD 2018 method is lower than *any one* in our generated sequence. Figure 9 presents three pairs where the generated adversarial examples by our HMC-based method all successfully fool the system as the target celebrities while Geekpwn CAAD 2018 method fails. Figure 10 shows a case that any one in the generated sequence by HMC-based method can fool the system with very high confidence while the adversarial examples

---

[6]https://clarifai.com/models/celebrity-image-recognition-model-e466caa0619f444ab97497640cefc4dc

generated by Geekpwn CAAD 2018 method being "caught" – which is recognized by the online systems and inferred to the true category.

We also list two failure cases in Figure 11. Note that the former one is slightly harder because the source and target are different races, but HMC-based method still generates few success samples. When looking at the big picture, these failure might be caused by our HMC-based method (or all the white-box adversarial attack methods) which are dependent of the targeted classifier. The decision boundary of different models may be slightly different from the global perspective, but the targeted attack forces the attacker to focus on the local decision boundary, which greatly magnifies this difference. It is worth noting that the phenomenon when using a long-run trajectory with Metropolis-Hastings correction which we mentioned in Appendix A.5 occurs again.

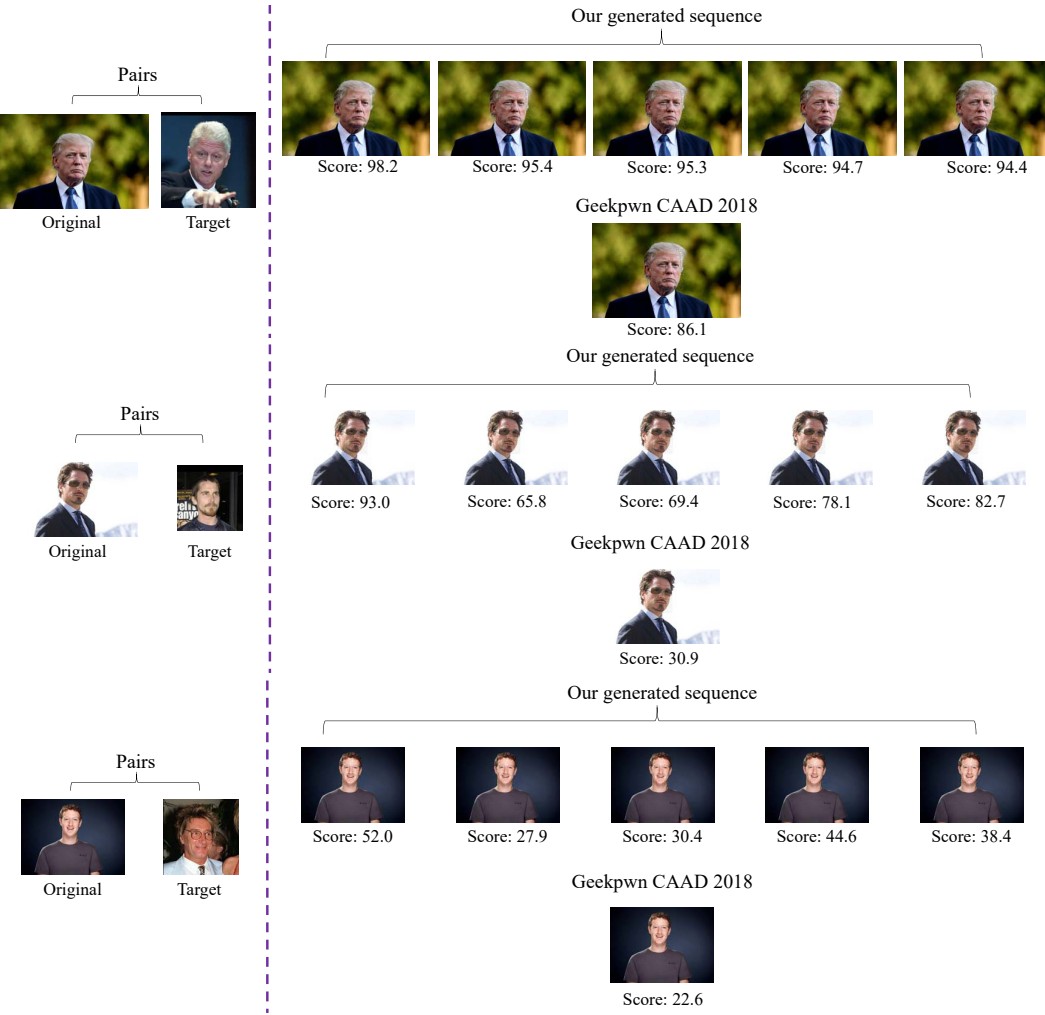

Figure 8: Success cases for both HMC-based method and Geekpwn CAAD 2018 method. The confidence score of adversarial examples generated by Geekpwn CAAD 2018 for each pair is lower than *any one* in our generated sequence.

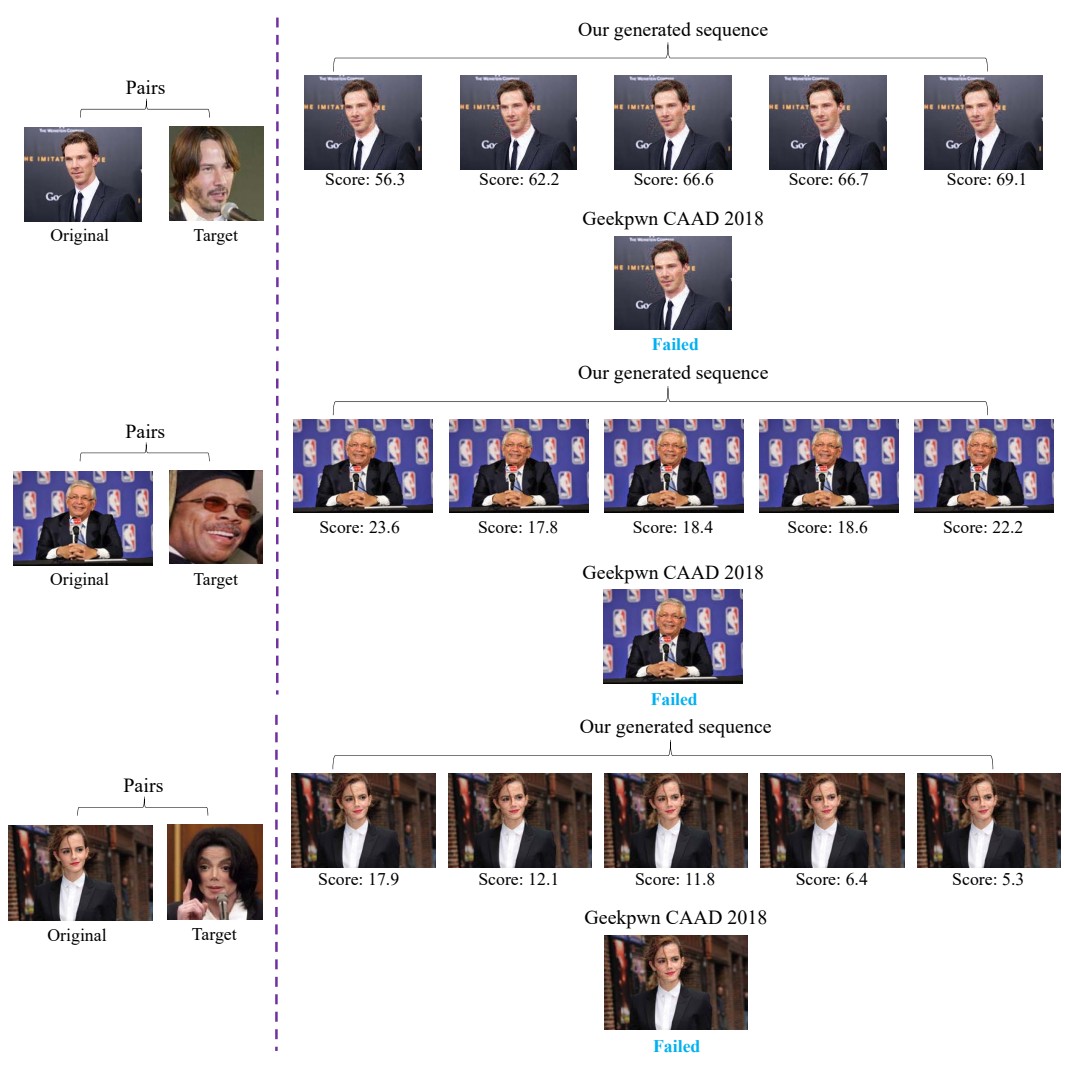

Figure 9: Three cases where our method succeeds but Geekpwn CAAD 2018 method fails

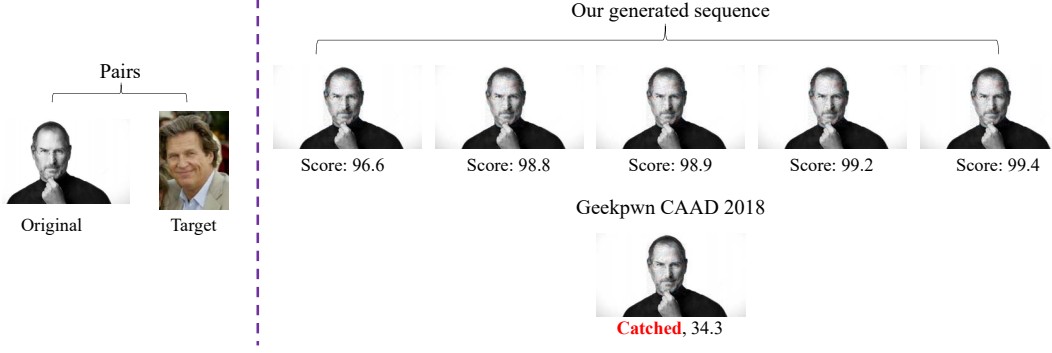

Figure 10: A case shows that an adversarial example generated by Geekpwn CAAD 2018 method is classified correctly by the online systems but any one in our generated sequence still fool the systems with high confidence score.

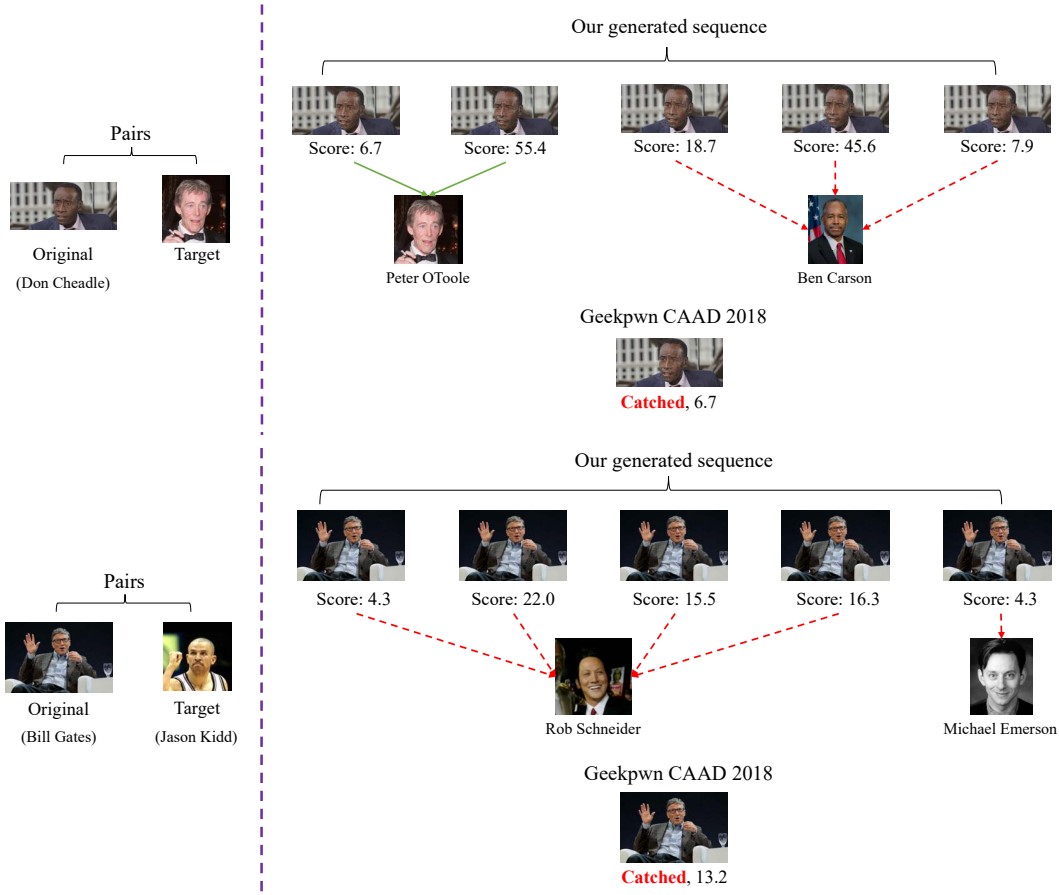

Figure 11: Typical failure cases of our method. In the task of attacking Don Cheadle as Peter O'Toole (the first row), HMC-based method only has a partial success due to their different colors. HMC-based method also fails in attacking Bill Gates as Jason Kidd (the second row). We believe that Bill Gates is an easy case for mutiple classifiers since there exist numerous related pictures about him and the recognition system will give priority to the correctness of him.

