# OpenReview forum: "Adversarial Training using Contrastive Divergence"
_ICLR.cc/2021/Conference — Reject_

### Official Review · AnonReviewer2 · 2020-10-25
**Contrastive divergence for adversarial sample generation. Theory formulation and results reporting**

**Rating:** 5
**Confidence:** 3

**Review:**

Adversarial examples are time-consuming to generate. In this paper, the adversarial training is reformulated as a combination of stationary distribution exploring, sampling, and training. A Hamiltonian system is proposed to model data samples from their initial states, and is shown as the general form of FGSM. The sample generation method is proposed via contrastive divergence with few training iterations. Experiments have been validated on datasets.

The formulation of HMC sounds interesting and its relationship towards FGSM/PGD is discussed. There are few issues towards the theoretical discovery and experimental validations:

1. The FGSM/PGD based attacks are formulated as the degeneration of HMC, as explained in Sec. 4.2. However, the experiments on the benchmarks consistently indicate PGD performs better than the proposed method ATCD. As ATCD utilizes a more advanced HMC, there lacks explanations of why the results do not correspond to the theory.

2. The proposed ATCD is claimed to efficiently generate adversarial examples while the performance seems to suffer from limited iterations. Is it possible to increase the iterations of ATCD for performance improvement?

3. As the efficiency is claimed as a major contribution of ATCD, the computational complexity and time cost compared to other methods (e.g., FGSM, PGD) shall be reported as well.

---

> ### Author Response · Authors · 2020-11-23
> **Analyses of experimental results, lengths of trajectory, and time complexity**
>
> Thanks for your valuable feedback! We address them in detail as follows.
>
> Q1. Explain why PGD still performs better than the proposed method ATCD.
>
> A1. Thank you for pointing this out. Due to the word limits, here we just give some major reasons. First, ATCD is an anytime algorithm expected to find better and better solutions the longer it keeps running. Since samples from the proposal distribution are not accepted automatically as posterior samples, which is essentially different from PGD with random starts. Our ATCD reach our final judgments on the acceptance probability to stay in (and return large numbers of samples from) high-density regions of the candidate distribution, while only occasionally visiting low-density regions. If we take a longer time to collect more samples, ATCD can depict the underlying distribution of adversarial examples according to adversarial examples and their corresponding frequencies. But in this paper, we mainly focus on reaching a balance of performance and efficiency. Second, the selection of potential energy and kinetic energy function has a great influence on HMC to simulate the trajectories of adversarial examples, like $\theta$ and $v$ should be not highly dependent. One potential solution is to reduce the relevance between $v_t$ and the current position $\theta_{t-1}$. That makes the sequence of samples into an approximate Markov chain. Besides, a well-designed schedule like Robust-Overfitting may be also helpful.
>
> Q2. Is it possible to increase the iterations of ATCD for performance improvement?
>
> A2. Yes. We have investigated different lengths of trajectory in Appendix A.6 in the original paper, which proved using a long-run trajectory of HMC with M-H resampling can better approach equilibrium distribution. But in this paper we mainly focus on improving the efficiency of adversarial training, so we recommend the short-run trajectory increases in speed and maintains the suitable clean accuracy to be a worthwhile trade-off.
>
> Q3. Add time complexity analysis of ATCD.
>
> A3. Thanks for the reminder. The actual time cost has shown in the last columns of Table 1,2,3. As for the time complexity analysis, please refer to Appendix A.4 in the revised version.

---

### Official Review · AnonReviewer4 · 2020-10-28
**Interesting topic with weak experimental results**

**Rating:** 6
**Confidence:** 2

**Review:**

Summary:
This paper proposed a new adversarial attack method by using Markov chain Monte Carlo.  Based on this attack method, a new adversarial learning method called adversarial training by using Contrastive Divergence (ATCD) which approaches equilibrium distribution of adversarial examples with only a few iterations is performed. The experimental results demonstrated the effectiveness of ATCD.

Pros:
1. The method of generating adversarial examples is new and efficient.
2. Experimental results with comparisons on ImageNet, CIFAR and MNIST datasets show that ATCD achieves a good trade-off between efficiency and accuracy in adversarial training.

Cons:
1. There is a notable natural accuracy gap between ATCD and Madry's PGD method and Free-m.
Many papers were aware of overfitting in adversarial training such as 'Overfitting in adversarially robust deep learning'. Nature accuracy and robust accuracy are sometimes conflicted in the latter epochs. It seems like some plots like Figure 3 may answer the question, but Figure 3 only plots the first 40 iterations (I guess it means 'epochs' here), while the total epoch is 105.
So only observe the results of the last epoch may not fair since maybe different methods have different convergence rates.
2. Some hyperparameter settings and sensitive analyses like $\rho$ and $\lambda$ are missing.
3. Although, the experimental results show the efficiency of ATCD, the formal time complexity analysis of ATCD should be performed.

---

> ### Author Response · Authors · 2020-11-23
> **Adding hyperparameter settings, sensitive analyses, and time complexity analysis**
>
> Thanks for your positive and valuable comments.
>
> Q1. Concern about Figure 3.
>
> A1. Thanks for pointing out the typo in Figure 3! Actually, Figure 3 has shown the whole process of advanced fast adversarial training methods since the total epoch of three advanced acceleration methods is 40. And they cannot directly compare with PGD whose total epoch is 105 because they speed up adversarial training by decreasing data access times or full forward / backward propagations. As for the configurations, all of them follow the official code of YOPO (Free can be considered as a special case of YOPO), where the initial learning rate is set to 0.2, reduced by 10 times at epoch 30 and 36. Equipped with the same initial learning rate and the learning schedule, it is fair to say that our method performs better than others.
>
> Q2. Clarify some hyperparameter settings and add sensitive analyses of $\rho$ and $\lambda$.
>
> A2. Thanks for your suggestion. We have included more setting details and added sensitive analyses of $\rho$ and $\lambda$ in Appendix A.3.5.
>
> Q3. Add time complexity analysis of ATCD.
>
> A3. Thank you for the constructive suggestion for us to consider. We have added the time complexity analysis of ATCD and other methods in Appendix A.4.

---

### Official Review · AnonReviewer1 · 2020-10-29
**Official Blind Review #1**

**Rating:** 5
**Confidence:** 4

**Review:**

Summary:
In this paper, the authors reformulated the generation of adversarial examples as an MCMC process and present a new adversarial learning method called ATCD, which approaches the equilibrium distribution of adversarial examples with only a few iterations by building from small modifications of the standard Contrastive Divergence. Extensive results with comparisons on various datasets show that ATCD achieves a trade-off between efficiency and accuracy in adversarial training.

Comments:

1 . The proposed algorithm ATCD views the generation of adversarial examples as a sampling procedure. Specifically, it can be seen as performing HMC sampling. Also, it modifies the adversarial example generation objectives using a modified contrastive divergence objective. Compared with free-adversarial-training, which also utilize mini-batch replay and use the last iterate’s result as initialization, the difference only lies in the adversarial example generation objective and the extra noise brought by the HMC sampling procedure. However, it is still not clear to me what actually causes the performance improvement. The different objectives or the sampling noise? I would suggest the authors conduct ablation studies to find out this answer and it would be clearer for the readers to understand the true driving force for the proposed method.

2 . In Eq (13) what is Q0 and Q1? It is better to formally define them. The intuition for this objective is a bit confusing to me. For standard contrastive divergence, it is the same as measuring the difference between the output probability between init point and K-step updated point. Here since the equilibrium distribution is unknown (fixed but unknown right?), how to compute the objective here with rho and lambda parameters? And why using different rho and lambda helps?

3 . Notice that even adversarial training based algorithms could cause obfuscated gradient problem, therefore, it might be a good idea to further evaluate model robustness via totally gradient-free methods, such as hard-label attacks. I would suggest the authors to also evaluate using the following method:

“RayS: A Ray Searching Method for Hard-label Adversarial Attack” KDD (2020)

In order to make the experimental results more convincing.

4 . There are some other recent adversarial training methods that the authors might want to comment on

"Improving adversarial robustness requires revisiting misclassified examples." ICLR (2019).
"On the Convergence and Robustness of Adversarial Training." ICML (2019).

---

> ### Author Response · Authors · 2020-11-23
> **Adding a critical ablation study, some relevant works, and clarifying some points**
>
> Thank you so much for taking the time to review the paper and we appreciate the comments.
>
> Q1. Conduct ablation studies to find out what actually causes the performance improvement.
>
> A1. Thanks for your valuable suggestion. We have added the relevant ablation study in Appendix A.3.3. In short, both the generation objective and the M-H resampling bring performance improvement for DNN.
>
> Q2. "In Eq (13) what is $Q_0$ and $Q_1$? And why using different $\rho$ and $\lambda$ helps?"
>
> A2. Sorry for the confusion. $Q_0$ and $Q_1$ are the output vector of the learning model (with softmax operator in the top layer) given input images $x+v_0$ and $x+v_K$. As you said, since the equilibrium distribution and its output $Q^{\infty}$ are unknown, we use the ground truth hard-label to approximate the output $Q^{\infty}$ when $\rho\neq \lambda$. The reason $\rho$ and $\lambda$ are introduced here is that we would like to make the connection with the traditional adversarial example generation objective. In fact, the original generation objective is equal to ours essentially when $\rho \ll \lambda$ except the entropy terms of $Q_0$ and $Q_1$. Experimentally, the sensitive analyses of $\rho$ and $\lambda$ in Appendix A.3.5 show its stability.
>
> Q3. Evaluate model robustness via totally gradient-free methods, such as hard-label attacks like "RayS".
>
> A3. According to your valuable comments, we have added accuracy results under RayS attack in Table 2. The experimental results show the effectiveness of our method.
>
> Q4. Comment on some other recent adversarial training methods.
>
> A4. Thanks a lot for recommending us the reference. In the updated version of the text, we have added these two works in the related work section.

---

### Decision · Program_Chairs · 2021-01-07
**Final Decision**

**Decision:**

Reject

**Comment:**

The authors propose adversarial training using contrastive divergence based on ideas from Hybrid Monte Carlo methods.
On the positive side the shown experimental results are promising both in terms of robustness and efficiency. On the negative side the paper seems to be written in a hurry. At several places terms are not defined, not explained or important details (e.g. parameters of the attack algorithms) are missing.

Thus the paper is not ready for publication yet and below the bar for ICLR but I encourage the authors to submit a significantly revised version to another conference.

Details:
- in (7) N(x) is nowhere explained or defined, here also several threat models are introduced but later on only l_infty seems
to be used e.g in Algorithm 1 (should be clarified)
- as noted in the reviews the definition in (13) is based on quantities nowhere introduced
- the potential U is only defined in the Algorithm (but then the arguments do not match with the RHS)
- the kinetic energy in the algorithm is different from what has been used before
- the parameters for the attacks used are not reported
- why is AutoAttack not used for all datasets? They report 28% robust accuracy for the model trained by the Madry group
 (https://github.com/MadryLab/robustness) whereas in the present paper it is 35%.
- the scales of the plots should be chosen such that the curves can be distinguished